# Groundwater in Crisis? Addressing Groundwater Challenges in Michigan (USA) as a Template for the Great Lakes

Alan D. Steinman [1,*], Donald G. Uzarski [2], David P. Lusch [3], Carol Miller [4], Patrick Doran [5], Tom Zimnicki [6], Philip Chu [7], Jon Allan [8], Jeremiah Asher [3], John Bratton [9], Don Carpenter [10], Dave Dempsey [11], Chad Drummond [12], John Esch [13], Anne Garwood [14], Anna Harrison [2], Lawrence D. Lemke [2,15], Jim Nicholas [16], Wendy Ogilvie [17], Brendan O'Leary [4], Paul Sachs [18], Paul Seelbach [8], Teresa Seidel [14], Amanda Suchy [2] and John Yellich [19]

1   Annis Water Resources Institute, Grand Valley State University, Muskegon, MI 49441, USA
2   Institute for Great Lakes Research, Central Michigan University, Mount Pleasant, MI 48859, USA; uzars1dg@cmich.edu (D.G.U.); harri25a@cmich.edu (A.H.); lemke1ld@cmich.edu (L.D.L.); suchy1a@cmich.edu (A.S.)
3   Institute of Water Research, Michigan State University, East Lansing, MI 48823, USA; lusch@msu.edu (D.P.L.); asherjer@msu.edu (J.A.)
4   Civil and Environmental Engineering, Wayne State University, Detroit, MI 48202, USA; ab1421@wayne.edu (C.M.); ax9873@wayne.edu (B.O.)
5   The Nature Conservancy, Lansing, MI 48906, USA; pdoran@tnc.org
6   Environmental Stewardship Division, Michigan Department of Rural and Agricultural Development, Lansing, MI 48909, USA; zimnickit@michigan.gov
7   Great Lakes Environmental Research Laboratory, NOAA, Ann Arbor, MI 48108, USA; philip.chu@noaa.gov
8   School for Environment and Sustainability, University of Michigan, Ann Arbor, MI 48109, USA; allanjw@umich.edu (J.A.); seelbach@umich.edu (P.S.)
9   LimnoTech, Ann Arbor, MI 48108, USA; jbratton@limno.com
10  Department of Civil Engineering, Lawrence Tech University, Soutfield, MI 48220, USA; carpenter@ltu.edu
11  For Love of Water, Traverse City, MI 49684, USA; dave@flowforwater.org
12  Drummond-Carpenter, Orlando, FL 32801, USA; cdrummond@drummondcarpenter.com
13  Office of Geologic Survey, Michigan Department of Environment, Great Lakes, and Energy, Lansing, MI 48909, USA; eschj@michigan.gov
14  Water Resources Division, Michigan Department of Environment, Great Lakes, and Energy, Lansing, MI 48909, USA; garwooda@michigan.gov (A.G.); seidelt@michigan.gov (T.S.)
15  Department of Earth & Atmospheric Sciences, Central Michigan University, Mount Pleasant, MI 48859, USA
16  Nicholas-h2o, Mason City, MI 48854, USA; jim@nicholas-h2o.com
17  Grand Valley Metro Council, Grand Rapids, MI 49504, USA; wendy.ogilvie@gvmc.org
18  Ottawa County Department of Strategic Impact, West Olive, MI 49460, USA; psachs@miottawa.org
19  Michigan Geological Survey, Western Michigan University, Kalamazoo, MI 49008, USA; john.a.yellich@wmich.edu
*   Correspondence: steinmaa@gvsu.edu

**Abstract:** Groundwater historically has been a critical but understudied, underfunded, and underappreciated natural resource, although recent challenges associated with both groundwater quantity and quality have raised its profile. This is particularly true in the Laurentian Great Lakes (LGL) region, where the rich abundance of surface water results in the perception of an unlimited water supply but limited attention on groundwater resources. As a consequence, groundwater management recommendations in the LGL have been severely constrained by our lack of information. To address this information gap, a virtual summit was held in June 2021 that included invited participants from local, state, and federal government entities, universities, non-governmental organizations, and private firms in the region. Both technical (e.g., hydrologists, geologists, ecologists) and policy experts were included, and participants were assigned to an agricultural, urban, or coastal wetland breakout group in advance, based on their expertise. The overall goals of this groundwater summit were fourfold: (1) inventory the key (grand) challenges facing groundwater in Michigan; (2) identify the knowledge gaps and scientific needs, as well as policy recommendations, associated with these challenges; (3) construct a set of conceptual models that elucidate these challenges; and (4) develop

a list of (tractable) next steps that can be taken to address these challenges. Absent this type of information, the sustainability of this critical resource is imperiled.

**Keywords:** groundwater; Great Lakes; agriculture; irrigation; urban water; coastal wetlands

## 1. Introduction

Groundwater is one of the most important resources on the planet, serving as a source of drinking and irrigation water for billions of people [1,2]. However, overextraction can result in depletion and contamination [3–6], as well as impacts to lakes as reduced groundwater inflows result in warming lake temperatures that trigger algal blooms [7]. Given the global significance of groundwater, considerable attention is being paid to the need for sustainable management of this resource, including managed aquifer recharge [8], integrated management planning [9], and a greater emphasis on conservation [10].

Given the ubiquity and visibility of surface water throughout the Great Lakes region, groundwater historically has been an understudied, underfunded, and underappreciated natural resource. Recent concerns associated with both groundwater quantity [5] and quality [11] have raised the profile of groundwater, but our understanding of this resource still lags compared with our surface water knowledge. A recent United States Geological Survey (USGS)-led assessment of science needs in the Great Lakes basin stated "little to no groundwater-quantity or -quality information is available to help manage water availability. The extent to which groundwater quantity and quality affect the overall function of the Great Lakes system is currently unknown" [12].

Total groundwater withdrawal within the Great Lakes basin was approximately 1510 million gallons per day (MGD) about 25 years ago [13], and has increased considerably since then. In Michigan, 45% of its citizens are served by groundwater as their primary drinking water source. Total groundwater withdrawals from all sectors in Michigan alone average 541 MGD [14]. Currently, the largest usage of groundwater in Michigan is for public water supply (208.9 MGD), followed closely by irrigation (208.5 MGD), and then industry (85.8 MGD) and livestock (21.4 MGD) [14]. However, there is limited documentation of residential usage in high-growth counties that utilize wells.

Groundwater's role in the environment receives less attention than its role in drinking water supply, but it supplies an average of 67% of the discharge in the larger tributaries flowing into the Great Lakes [15] and provides cold, high-quality flow for highly valued trout streams in the region [16,17]. Estimates such as these are even more difficult to make for groundwater contributions to wetlands and inland lakes because of their dynamic nature and lack of monitoring data. Although there are some estimates of groundwater inputs to wetlands in the state [18,19], very few field studies have examined groundwater–wetland interactions in coastal areas of the Great Lakes [20]. Xu et al. [21] recently estimated that direct groundwater discharge accounts, on average, for a relatively small amount of positive basin supply in the Great Lakes (0.6–1.3%), although it is much more important in nearshore than offshore regions.

Not only are groundwater quantity and quality threatened by overextraction [9,22], but so are groundwater-dependent natural systems. In the Great Lakes, issues have arisen around both groundwater quantity and quality in recent decades [23,24]. Private sector withdrawals for bottled water have resulted in lawsuits [25] and tribal protests. These groundwater conflicts contributed to the development of a water withdrawal assessment tool in Michigan [26], and concerns over emerging contaminants such as PFAS, process water from hydraulic fracturing, and overall sustainability [27–29] are becoming more common. The Michigan Water Use Advisory Council, most recently codified in 2018 PA 509, provided a series of recommendations to advance and improve conservation, data collection, modeling, research, refinement, and administration of Michigan's water withdrawal

assessment process [30]. A Michigan Hydrologic Framework has been proposed to manage the state's water through the use of integrated hydrologic models, data, and analysis [31].

Given the increasing pressures being placed on groundwater in the Great Lakes region, despite the appearance of abundant freshwater supplies, a virtual summit was held on 3–4 June 2021 to address this apparent paradox and hone in on key groundwater issues in those sectors that are experiencing particular ecological stress or have broad application: agricultural systems, urban (developed) systems, and coastal wetlands. We focused specifically on Michigan, which is in the midst of several critical groundwater-related issues, although we recognize that these issues are germane to the entire Great Lakes basin and beyond, and our approach was designed to be scalable and transferable. We also developed conceptual models (see below) for each of these sectors, which helped identify information needs and gaps, and allowed the groundwater issues in each sector to be readily compared with one another.

Experts from the academic, private, and public sectors were invited (see Table S1) and charged with two tasks: (1) inventory the groundwater challenges in urban (developed), agricultural, and coastal wetland ecosystems; and (2) frame those challenges in a hierarchical Driver-Pressure-State-Impact-Response (DPSIR) model, as recommended in the CIGLR-funded Conceptual Frameworks summit [32].

The overall goals of this groundwater summit were fourfold: (1) inventory the key (grand) challenges facing groundwater in Michigan; (2) identify the knowledge gaps and scientific needs, as well as policy recommendations, associated with these challenges; (3) construct a set of conceptual models that elucidate these challenges; and (4) develop a list of (tractable) next steps that can be taken to address these challenges.

This paper is structured to first describe the summit structure and process, followed by the summit findings that are subdivided into the following categories: general, agricultural-sector; urban sector; and coastal wetland sector. Each of the three sectors has a consistent structure consisting of key challenges, DPSIR models for each of the challenges, and a discussion section containing actionable recommendations. We conclude with a summary section containing study limitations and our overall recommendations.

## 2. Summit Description and Methodology

The summit was funded in 2019 by the Cooperative Institute for Great Lakes Research (CIGLR), one of 16 NOAA-sponsored Cooperative Institutes throughout the USA. A seven-person steering committee developed the format and overall approach for the summit. The invited participants were chosen to represent a variety of disciplines within the groundwater sector and included representatives from five government entities, six universities, three non-governmental organizations, and two private firms. The steering committee members were intentional in inviting both technical (e.g., hydrologists, geologists, ecologists) and policy experts to the summit. The original intent was to hold the two-day summit on the campus of the University of Michigan in June 2020, but COVID-19 disrupted that plan, and instead, the summit was held virtually on 3–4 June 2021. LimnoTech provided technical support, allowing participants to enter their findings in Google Docs during each breakout session. Participants were assigned to the agricultural, urban, or coastal wetland breakout groups in advance, based on their expertise (Table S1). In addition, the steering committee developed expectations that were explicitly identified and conveyed to participants beforehand. Each breakout group identified individuals to take notes and report out on their findings. Steering committee members were assigned to the breakout groups to facilitate discussion and keep conversations focused and on topic.

Day 1 (Table S2) included a brief overview of the summit and CIGLR by the CIGLR interim director Tom Johengen, followed by an overview of the summit format and expectations by Al Steinman (GVSU). Teresa Seidel (EGLE) provided her perspective on the state of groundwater in Michigan, noting that: groundwater is a forgotten stepchild in MI environmental programs and needs better integration; the water conservation message has been lost; and that we need a state policy framework for groundwater and reinvestment in

infrastructure. This was followed by a whole group discussion facilitated by Don Uzarski (CMU) that inventoried the key groundwater challenges facing Michigan. The participants then entered into three virtual breakout rooms, where they first prioritized the challenges in their respective sectors, followed by the construction of conceptual models using the DPSIR (Driver-Pressure-State-Impact-Response) framework (see below), and finally, they identified the science and policy gaps/recommendations for these key challenges. The entire group reconvened in the last session to discuss these challenges and recommendations.

One of the unique features of this summit was the use of a conceptual model framework to provide structure and consistency among the three groundwater sectors. We built upon the CIGLR-funded 2018 conceptual frameworks summit, which identified the DPSIR model as the most appropriate conceptual model for describing and visualizing how the Great Lakes are structured and their component parts interact with each other [24]. The DPSIR framework examines key relationships between society and the environment (Figure 1), and, therefore, can be useful for structuring and communicating policy-relevant research about environmental issues [33].

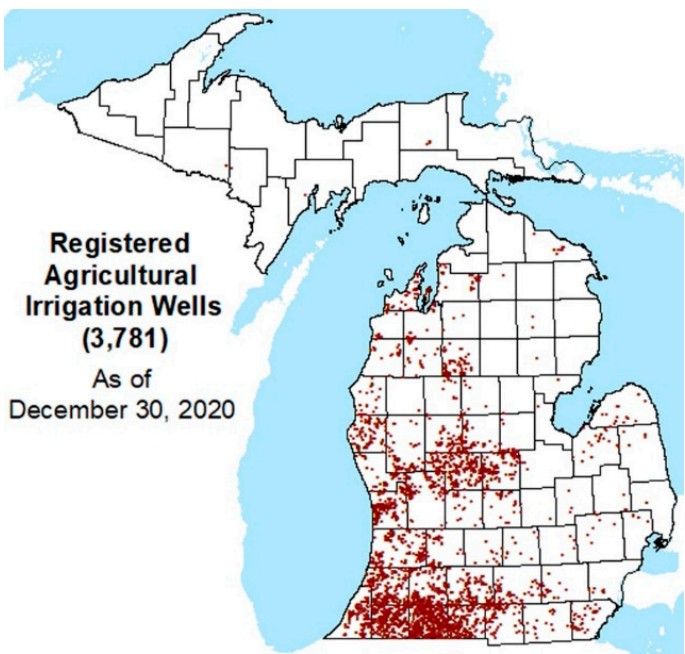

**Figure 1.** Registered agricultural irrigation wells in Michigan, as of 30 December 2020 (EGLE 2021).

### 3. Results

*3.1. General Findings*

Discussions on day 1, during both breakout groups and entire assemblage sessions, revealed a number of cross-cutting topics that applied to all three groundwater sectors and had broad relevance. These topics were divided into Scientific (technical) and Management-oriented (non-technical) categories, and are briefly discussed below. They were followed by the key challenges and DPSIR models for the agriculture, urban, and coastal wetland sectors.

3.1.1. Scientific Issues

- Groundwater budgets: there is a pressing need for better information on aquifer recharge and withdrawals throughout Michigan, but especially in regions where groundwater pressures already exist or are anticipated to soon become worse.
- Contamination: this concern has been highlighted and exacerbated recently by the discovery of PFAS in many groundwater systems, but it has been an issue for decades with other pollutants such as excess nitrate from fertilizer applications, excess phosphorus from septage, and trichloroethylene from manufacturing processes [34], as

well as other human-produced contaminants from commercial and manufacturing operations and processes.

- Forecasting: although we must gain a better understanding of the current state of Michigan's groundwater, we also need to envision the future state of supply and demand. This has multiple considerations, including, but certainly not limited to, the potential impacts of climate, land use/cover, and demographic shifts that may change withdrawal and recharge rates [35,36]. These factors can influence and exacerbate the movement of pollutants lurking in the groundwater (those already known), as well as those not yet discovered.

- Connectivity: the notion of hydrologic connectivity was a consistent thread in our discussions with respect to both surface water and groundwater. Concerns were expressed about the public's general lack of understanding of this concept, as well as the lack of geological information regarding connectivity because of limited three-dimensional (3D) geologic mapping in many areas.

- Information Tools and Gaps: these two issues are related, as we have significant information gaps on the 3D extent of glacial aquifers, aquifer water budgets (see above), and groundwater quality, but conveying this complex, technical information in an intuitive and easily digestible manner is equally difficult. Increased efforts to complete 3D mapping of the geologic substrata, as well as other visualization tools will allow us to share complex information efficiently, display information effectively, and communicate the information intuitively. Although better information and science is a critical step forward, it does little good without effective decision-support systems and information/visualization tools.

### 3.1.2. Management-Oriented Issues

- Public Education: anecdotal evidence suggests there is a substantial portion of society that perceives groundwater as vast pools of "underground lakes and rivers"; there is a pressing need to better educate the public, including elected officials, on groundwater science. The W.K. Kellogg Foundation, in association with the Institute of Water Research at Michigan State University, developed the Groundwater Education in Michigan (GEM) Program in 1987. This program demonstrated that successful source water protection programs must be persistent and depend upon strategic partnerships among federal and state agencies, universities, local and district health departments, watershed groups, conservation districts, and others. Similar efforts need to be reconstituted and maintained into the future.

- Water Use Conservation: how do you convince the Michigan public to conserve water, whether it be from the surface or ground, when they are surrounded by four of the largest lakes on the planet and live in a state with over 10,000 inland lakes? This conundrum was termed the "fallacy of universal ubiquity" by one of the summit participants.

- Land and Water Management: although "conservation" frequently refers to reductions in the use of groundwater, the term may apply also to practices that benefit keeping or maintaining groundwater in the system, including multiple agricultural and urban best management practices (cover crops, green infrastructure), as well as legal mechanisms that restrict development or land use change in high groundwater recharge areas. The benefits of such practices must continually be documented, and subsequently, incentives for implementation will need to be established.

- Environmental Justice: there was an acknowledgment that important segments of our society were not represented at the summit, including representatives from BIPOC communities. Clearly, this limits the scope of our findings and recommendations but highlights that additional efforts, strategies, and capacities are needed to engage with, and understand, this issue from multiple perspectives.

- Advocacy: considerable discussion was devoted to the need to lobby more effectively on behalf of groundwater. This "Sixth Great Lake" [37] deserves increased attention, but there was no clear consensus on how this should be accomplished, especially given

the mix of NGO, academic, and government actors at the summit. Each of these groups has perspectives that in some way must comport with their institutions' guidelines and codes of conduct. However, there was general agreement regarding the need for more effective strategies to garner the resources and attention on groundwater as a growing Great Lakes issue. A few of the ideas that were discussed included: (1) using the GLRI (Great Lakes Restoration Initiative) to create a new Focus Area devoted to groundwater or the Great Lakes Water Quality Annex (GLWQA) Annex 8 update; (2) an annual MI conference devoted to groundwater (although concerns were expressed about preaching to the choir); (3) conducting a study estimating the economic value derived from groundwater use in the state through the agricultural, manufacturing, drinking water, etc. sectors; and (4) utilizing the Water Use Advisory Council as a vehicle for greater advocacy.

### 3.2. Groundwater in the Agricultural Sector

Agricultural use of groundwater is increasing in Michigan, with an estimated total of nearly 10,000 agricultural irrigation wells in the state [38], over one-third of which were installed in the past 10 years. The food and agriculture industry is a critical part of Michigan's economy, contributing an estimated $104.7 billion annually to the state's economy and employing 923,000 Michiganders—22% of the state's workforce [39]. Irrigation water supply is critical to maintaining and enhancing that economic flow, yet concerns over groundwater quantity and quality continue to escalate.

### 3.2.1. Key Challenges

Three key challenges were identified by the agriculture work group:

(1) The increasing use of groundwater for agricultural irrigation—the need to irrigate, primarily using groundwater sources, has dramatically increased in Michigan over the last two decades. Between 1997 and 2017, the amount of irrigated cropland in Michigan expanded by ~51,175 ha—a 64.6% increase [40,41]. In the period 2008–2020, the number of agricultural irrigation wells in Michigan more than doubled, increasing by 152% [38,42,43]. Over 3600 high-capacity agricultural irrigation wells have been developed in Michigan over the past decade (Figure 1). The irrigation sector (dominated by agriculture) withdrew an average of 154.3 MGD (million gallons per day) of groundwater in 2010 and 208.5 MGD in 2019 [14,44], and questions are being asked about sustainability [45].

(2) The increasing contamination of groundwater from agricultural nutrients and chemicals—fertilizer use in Michigan increased steadily from the 1930s, when commercial fertilizers first became available, to the early 2000s when total consumption of fertilizers in Michigan leveled off [46]. According to USEPA [47], the amount of N fertilizer purchased in Michigan in 2007 contained 243.6 million kg of N. The longer-term trend shows an 8% decrease in N fertilizer sales in Michigan, comparing 2002–2006 with 2007–2011 [47]. Virtually all agricultural commodities produced in Michigan require treatment with pesticides to prevent serious yield losses from disease and insect, nematode, vertebrate, or weed pests [48].

(3) Groundwater Recharge and Drainage Best Management Practices (BMPs)—in general, agricultural producers deal with excessive soil moisture nine months out of a year, with the remaining three months committed to irrigating crops during periods of limited precipitation. Although traditional practices of subsurface drainage have proven successful in reducing excessive soil moisture, thereby creating optimal conditions for crop production, a detrimental impact of such practices is a decrease in groundwater recharge [49]. Subsurface drainage systems, in general, transport surplus water in the soil's root zone to surface drainage ditches, and ultimately into rivers and lakes. Over time, this removal of water from agricultural fields negatively impacts localized groundwater recharge rates, resulting in a decline in available groundwater from shallow aquifers, and also contributes to downstream eutrophication [50,51].

### 3.2.2. DPSIR Models

Groundwater and Irrigation Model

The main driver in this model (Figure 2) is the contract-grower nature of agriculture in parts of Michigan (e.g., seed corn production in SW Michigan), combined with the increased importance to irrigate other high-value crops to mitigate the increasingly volatile climate risk. SW Michigan is a well-known specialty crop production region where all of the seed corn and chipping potato fields are irrigated, as are the fields of snap beans, tomatoes, pickling cucumbers, peppers, and summer squash. The farm gate value of the seed corn and chipping potatoes industries in Michigan was over $100 million and about $33 million, respectively, in 2014 [52]. The combined farm gate value of the other specialty crops was about $74 million in 2014 [52]. Blueberry production is also concentrated in SW Michigan, contributing over $120 million in farm gate receipts to the local economy annually. About 79% of Michigan's blueberry acreage is irrigated [52].

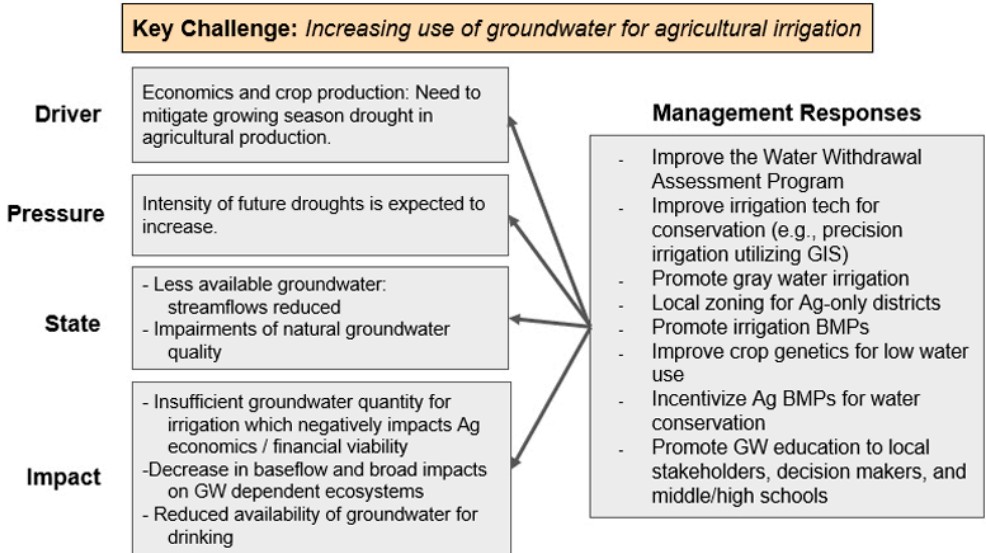

**Figure 2.** Groundwater and Irrigation Model.

The pressure in this model comes from climate change. Although the average annual precipitation in Michigan has increased by almost 10% since 1901, the change in the seasonal distribution of precipitation is the primary stressor of rainfed agriculture [53]. Heat waves and droughts have become more frequent and more intense since the 1960s [54]. Future growing-season precipitation is predicted to increase in the short-term, but decrease by 5–15% by the end of the century [55]. The effect of climate change on groundwater in the Great Lakes basin has a high degree of uncertainty, with high spatial and temporal variability across the region [35].

Due to current and future increases in water use, the state has changed as well, with less high-quality groundwater available for all uses. Groundwater conflicts are anticipated to continue growing. The coupled nature of groundwater and surface water means that the increased use of groundwater will lead to reduced stream flows, especially in the southwest and west-central regions of the lower peninsula of Michigan. Natural concentrations of dissolved chloride in most shallow aquifers in the Great Lakes Region are typically less than 15 mg/L [56]. Curtis et al. [57] identified four regions in Lower Michigan that stand out as statewide hotspots with elevated (>20 mg/L) or severely elevated (>250 mg/L) chloride concentrations in groundwater, including parts of Arenac, Bay, Huron, Iosco, Kent, Lenawee, Midland, Ottawa, Saginaw, Sanilac, St. Clair, and Tuscola Counties. They [57] documented that chloride concentrations in groundwater in the regional discharge zones of Lower Michigan are consistently and significantly higher than those in recharge zones. Within local hotspots, they concluded that the relative impact of upwelling brines was

controlled by (1) large-order streams promoting the natural upwelling of deeper (more mineralized) groundwater to the surface; (2) the occurrence of low permeability sediments at or near the land surface that restrict fresh water recharge of deeper groundwater-bearing zones; and (3) the space–time evolution of residential well withdrawals, which pump year-round and induce a slow migration of saline groundwater from its natural course.

These changes have resulted in severe impacts, such as insufficient groundwater quantity (and in some areas, quality) for irrigation and residential use, which threatens the financial viability (and potentially, the social stability of groundwater-dependent areas) of high-value specialty crop agriculture in Lower Michigan. The dramatic increase in groundwater uses has already decreased the baseflow in some streams and negatively impacted a variety of groundwater-dependent ecosystems. The expanding uses of groundwater and the concomitant rise in irrigation shunting water out of infiltration and system recharge in some regions of Michigan has both reduced the availability of potable water for drinking and irrigation, and continues to increase the potential for conflicts over groundwater availability.

Although the irrigation issue is significant, several tractable management responses can be implemented to address the problem:

- Improve the Michigan Water Use Program and the Water Withdrawal Assessment Tool by funding and implementing the recommendations of the Water Use Advisory Council [30]; see below.
- Improve the efficiency of low-loss irrigation technology and conservation measures.
- Promote precision irrigation technologies utilizing GIS and in-situ monitoring.
- Advocate for gray water irrigation technologies and adoption.
- Improve local zoning by adopting "ag only" zones and open space uses of regional groundwater recharge areas.
- Incentivize the widespread adoption of irrigation BMPs [58].
- Advocate for enhanced research funding of drought-tolerant/low-water-use crop genetics.
- Promote and sustainably fund groundwater education to local stakeholders, decision-makers, and middle/high school students.

Groundwater Contamination Model

The focal driver of this model (Figure 3) is related to the dependence of intensive agriculture on nutrient and pest management practices. Michigan State University Extension (MSUE) recommends fertilizing most field crops and vegetables in Michigan [59,60] based on soil fertility tests, soil texture, crop type, and realistic yield goals that are achievable at least 50% of the time. MSUE also recommends an integrated pest management approach using a combination of techniques, including cultural methods and herbicides [61].

This results in the pressure of fertilizer application to replace the nutrients that crops remove from the soil. It is estimated that average corn yields would decline by 40% without nitrogen (N) fertilizer and even greater declines would occur if other macronutrients, such as phosphorus (P) and potassium (K), were also limited [62]. The drivers also result in an additional pressure of chemical applications to control weeds and pests. Weeds cause tremendous losses in crop yield and quality. Based on data from 2007–2013 for corn and soybean, 2007–2016 for dry bean, and 2002–2017 for sugar beet, the average percent yield losses with no weed control were: 52% in corn; 49.5% in soybean; 71.4% in dry bean; and 70% in sugar beet [63].

Fertilizer applications have resulted in elevated nitrate concentrations that have contaminated groundwater in Michigan and across North America [64–66]. Between 2007 and 2017, the State of Michigan Drinking Water Laboratory tested for nitrate in 78,826 samples of drinking water and reported that about 19% of the samples had detectable levels of nitrate, 3% had elevated levels of nitrate (i.e., 5–10 mg/L), and about 1.8% exceeded the drinking water standard of 10 mg/L [67]. Agricultural sources of nitrate include wastes from livestock operations and farm fertilizers. The Michigan Department of Agriculture and Rural Development (MDARD) Water Monitoring Program routinely tests the water

quality of privately owned water wells and has found one or more pesticides in 2.3% of the wells they tested [68].

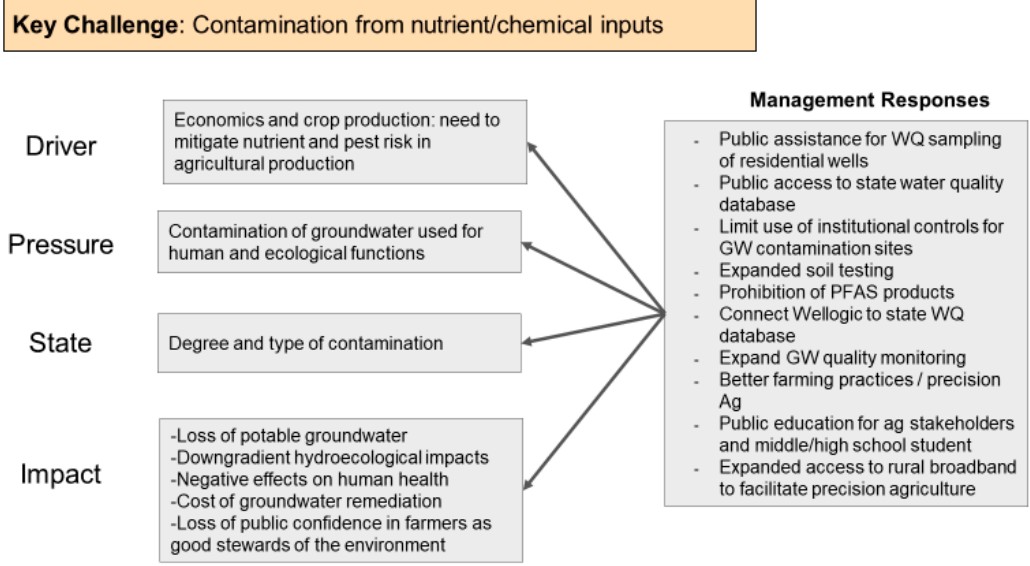

**Figure 3.** Groundwater Contamination Model.

The impact of elevated nutrients and chemicals in groundwater is their potential threat to human health. Nitrate ($NO_3$) is a form of nitrogen that is chemically reduced to become nitrite ($NO_2$). Nitrate in drinking water can cause methemoglobinemia, a blood disorder primarily affecting infants under six months of age. Some studies suggest that exposure to nitrates and nitrites by pregnant women may increase the risk of complications such as anemia, spontaneous abortions, premature labor, or preeclampsia.

Epidemiologic data suggest an association between various birth defects in offspring and the maternal ingestion of nitrate from drinking water [69]. Other studies suggest an increased risk of contracting leukemia, lymphoma, and brain, kidney, breast, prostate, pancreas, liver, lung, and skin cancers due to pesticide exposure [70]. According to FLOW [67], a 2008 Minnesota study found that well owners whose groundwater nitrate levels exceeded 10 mg/L typically paid nearly $2000 for a treatment system or more than $7000 to replace their well. In addition, in some cases, transport of nutrients and chemicals from groundwater to surface waters can be accelerated so this contamination can be bidirectional.

There is a clear need for management responses to this issue; Michigan currently lacks a coherent groundwater policy and law that reaffirms groundwater is part of a single hydrologic cycle, and that protecting this public-trust resource from impairment and degradation is paramount. As the State of Michigan's 30-Year Water Strategy observes, "Groundwater use and value is increasing, and the state must invest in the information and decision systems to realize groundwater's full value, promote its wise use and protect its hydrological and ecological integrity" [71]. The Michigan Legislature should appropriate funding to assist owners of residential drinking water wells to obtain partial chemistry, bacteriological, and arsenic tests of their well water and make these data available to the public in a geospatial format. Addressing the input side, the State of Michigan, environmental NGOs, and the private sector should aggressively endorse and financially enhance the Michigan Agriculture Environmental Assurance Program and similar voluntary programs [72], which help farmers adopt rigorous BMPs for nutrients, animal wastes, and pesticides, thus protecting both surface water and groundwater. Other management responses to this challenge include limiting the use of institutional controls especially for industrial contamination, expanding soil testing in agriculture, prohibiting PFAS products, connecting the Wellogic database to the state water quality database, expanding groundwa-

ter quality monitoring, promoting and funding public education about groundwater, and expanding rural broadband access in order to promote precision agriculture.

Groundwater Recharge and Drainage Best Management Practices (BMPs)

The key driver of the agricultural drainage BMP model (Figure 4) is related to climate change. Since 1951, total annual precipitation has increased by 14% across the Great Lakes states [73]. Projections indicate notable increases in precipitation, as much as 30% on average, into the near future. Increases in global temperature averages are directly correlated to more frequent and significant precipitation events. As the Earth's temperature warms, it results in increased water vapor capacity in the atmosphere, which ultimately turns into precipitation. Increasing temperatures also contribute to warmer surface temperatures of the Great Lakes thereby reducing winter ice cover. Diminished ice cover promotes more lake-effect snow precipitation. These environmental factors ultimately lead to wetter springs, which necessitate that agricultural producers remove excessive moisture from their fields to ensure successful crop yields [74]. Subsurface drainage of cropland improves productivity and yield consistency, which translates into greater financial returns.

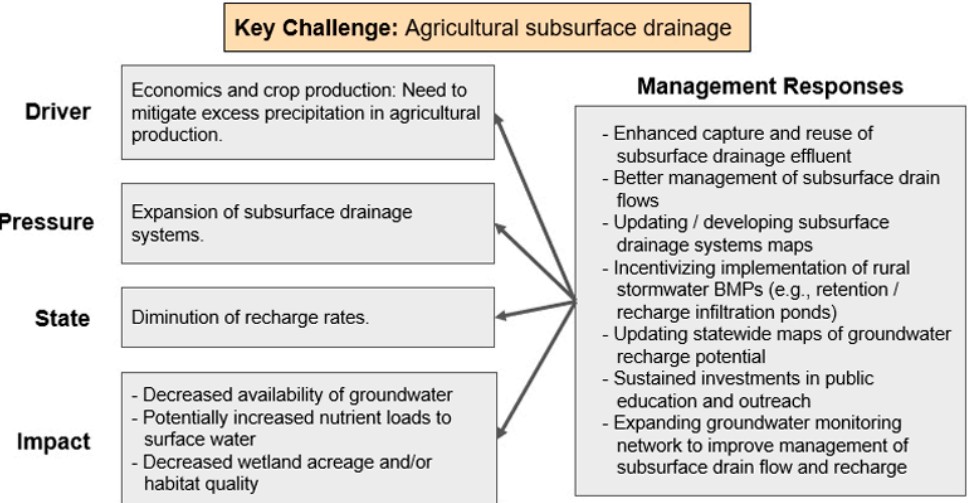

**Figure 4.** Groundwater recharge and drainage best management practices model.

In order to ensure successful crop yields, and to maintain economic stability amidst the many environmental variables that impact agricultural operations, producers must deal with the pressure of adapting and managing their fields—which in this instance necessitates the installation of subsurface drainage systems to remove excess soil moisture in the root zone. Due to consistent increases in precipitation, agricultural producers have installed more extensive subsurface drainage systems. In 2017, 38% of cropland in Michigan was drained by subsurface systems [75]. Between 2012 and 2017, subsurface drainage of cropland increased by more than 4.5%. Based on continued increases in precipitation across the Great Lakes, more and more agricultural producers are installing subsurface drainage systems, which may have negative water quality impacts on downstream receiving waters [50,76].

These pressures have resulted in a state of declining availability of high-quality groundwater. One of Michigan's primary bedrock aquifer systems, the Marshall Sandstone, is a primary source of water for irrigation and domestic purposes in many counties. Over the last 40 years, as a result of continued groundwater withdrawals from the Marshall Aquifer, static water levels (SWL) in Ottawa County (MI) have dropped more than 13 m, with an additional 6+ m drop likely within the next 15 years if withdrawal rates continue [77]. A significant contributing factor to the SWL decline is not simply withdrawal rates, but a dramatic lack of groundwater recharge into the Marshall Aquifer system. In a sustainable system, groundwater recharge rates should exceed groundwater withdrawal rates. In the

case of Ottawa County, a continuous clay layer in the shallow subsurface covers a large portion of the county that restricts freshwater recharge to the underlying bedrock layer.

In those areas where freshwater recharge can infiltrate into the bedrock aquifer system, subsurface drainage systems may severely limit this recharge. The transport of excess water away from agricultural areas and into drainage ditches and ultimately downstream into wetlands, rivers, and lakes also contributes to flooding challenges in low-lying areas, along with increased potential for excessive nutrient loads in those surface water systems.

Management responses to this problem include: (1) enhancing the capture and reuse of subsurface drainage effluent; (2) better management of subsurface drainage flows; (3) updating and developing drainage systems maps; (4) incentivizing the implementation of rural stormwater BMPs (e.g., retention/recharge infiltration ponds); (5) updating statewide maps of groundwater recharge potential; (6) improving public education and outreach; and (7) expanding groundwater monitoring networks to improve management of subsurface drain flow and recharge.

### 3.2.3. Discussion

The agriculture breakout group agreed on four major challenges applicable broadly across each of the models, including: (1) a trend towards increasing irrigation for agricultural uses; (2) contamination of groundwater from agricultural nutrient and chemical inputs; (3) an increase in agricultural best management practices that benefit groundwater recharge; and, (4) the need for approaches and models to determine groundwater availability, especially in response to climate change.

To make progress on addressing these challenges, the breakout group identified multiple actionable steps related to policy and practice, science and infrastructure, and education and outreach. Importantly, the breakout group highlighted the foundational work of the Michigan Water Use Advisory Council [30]. The following activities are not exhaustive but will be essential to the long-term management of groundwater in agricultural settings.

Policy and Practice

- Develop techniques to recycle/reuse drain tile water and residential gray water;
- Advance precision agriculture, including the enabling conditions such as expanding soil testing and expanding access to broadband;
- Assess local and state ordinances as well as regional planning efforts that protect and conserve groundwater;
- Employ new approaches to improving irrigation efficiencies; and
- Connect Wellogic to the state water quality database to increase consistent and accessible data.

Science and Infrastructure

- Assess groundwater connectivity, with a focus on movement of groundwater from one system to another;
- Update statewide groundwater recharge maps;
- Develop and/or update tile drain maps;
- Assess groundwater changes via techniques such as calibrating GRACE (https://grace.jpl.nasa.gov/applications/groundwater/ (accessed on 4 February 2022)) to the Great Lakes region;
- Invest in core development of precision irrigation, broadband availability, real time collaborative monitoring networks, and use of satellite imagery to guide agricultural practices; and
- Assess the benefit of agricultural best management practices for groundwater quality and quantity.

Education and Outreach

- Continued communication with and among the agricultural community on groundwater issues, especially utilizing farmer-led watershed groups;
- General education with the public (i.e., where does your water come from?);
- Training for water well drillers for consistency to improve the accuracy of lithology data;
- Improve/develop new water conservation programming through existing programs like MAEAP or others; and
- Stronger and more intentional engagement between various governmental, academic, NGO and business communities.

### 3.3. Groundwater in the Urban Sector

Urbanization presents many challenges to groundwater management. The built environment introduces complex flow pathways for groundwater and its associated contaminants in both lateral and vertical directions. Many urban centers in the Great Lakes basin rely on the abundant surface water for drinking water, industrial uses, and residential irrigation; thus, there is often less emphasis on the groundwater resources in these urban places compared with their rural counterparts. One result is fewer groundwater monitoring installations in the urban centers, which in turn limits our knowledge of groundwater characteristics within the built environment. This constrains our ability to manage urban groundwater resources and generate robust sustainability plans for urban centers in the future.

Michigan's geology is the template for urban groundwater, but anthropogenic features heavily impact the groundwater hydrology in these areas; these features include subsurface infrastructure (sewerage, drinking water, gas, and other utilities), structural foundations, infill soils, rerouting of rivers, and legacy contaminants [78–80]. These can lead to changes in groundwater quality, quantity, and spatial disposition. A common example is the leakage/exchange that occurs between groundwater and sewer pipes. Due to the elevated water tables in many Great Lakes urban centers, green stormwater infrastructure (GSI) suitability issues are a concern [81,82]. On the regional scale, leaky distribution systems and highway dewatering systems can impact groundwater flow throughout urban centers [83]. The American Society of Civil Engineers 2017 Infrastructure Report Card indicates that wastewater infrastructure (e.g., sewer pipes) is deteriorating, in poor condition, and leaking; many Great Lakes states ranged from C (mediocre) to D minus (poor) on the report card [84]. The lack of management oversight of urban groundwater raises concerns of potential health impacts from vapor intrusion, subsurface seepage into homes, interconnections of sewer/water lines, and impact to surface water resources.

### 3.3.1. Key Challenges

The urban groundwater breakout group identified three key challenges:

1. Presence of anthropogenic contaminant sources
2. Elevated and fluctuating groundwater tables
3. Anthropogenic modifications to urban groundwater systems

Presence of anthropogenic contaminant sources—urban areas typically have a high concentration of commercial and industrial uses and a legacy of brownfield locations. Anthropogenic contaminant chemicals of concern include, but are not limited to, forever chemicals (e.g., Per- and polyfluoroalkyl substances), chlorides, and hydrocarbons [85]. Chemical interactions with water can lead to the transport of these chemicals around urban centers generating health concerns, such as vapor intrusion into residential and commercial structures [86].

Elevated and fluctuating groundwater tables—many urban centers in the Great Lakes basin are in shallow groundwater zones. Effects of climate change, at both the global and the local scales, present unique groundwater management concerns [87–89].

Anthropogenic modifications to urban groundwater systems—urban areas are zones of high anthropogenic disturbance that alter the natural water cycle. Urban water cycles require new and diverse strategies to adequately manage groundwater depth, flow, and quality [90,91].

### 3.3.2. DPSIR Models

Three models were created from the three key challenges identified during the preliminary scoping of urban groundwater concerns. Two of the models, the Groundwater Table Model and the Anthropogenic Modifications Model, focused on groundwater flow issues and associated concerns. Hence, both models have overlapping management responses. The third model, the Anthropogenic Contaminant Sources Model, focused on chemical releases and the role of urban groundwater in the fate and transport of chemicals. Although there were some overlapping management themes, the management responses in this model differed from the other two models.

Anthropogenic Contaminant Sources Model

The two drivers of anthropogenic contaminant sources (Figure 5) are urban/industrial economic development and land use. The pressure is the inappropriate release of toxic chemicals into the environment. The state of many urban places is spatial clusters of subsurface chemical releases [92]. This has left many urban locations with contaminated soils, groundwater, and in some cases, impacted surface water. Sites of known contamination are often associated with vulnerable populations [93–95]. The impacts of anthropogenic chemical releases include beneficial and ecological use impairments, degraded human and ecological health, and adverse economic consequences.

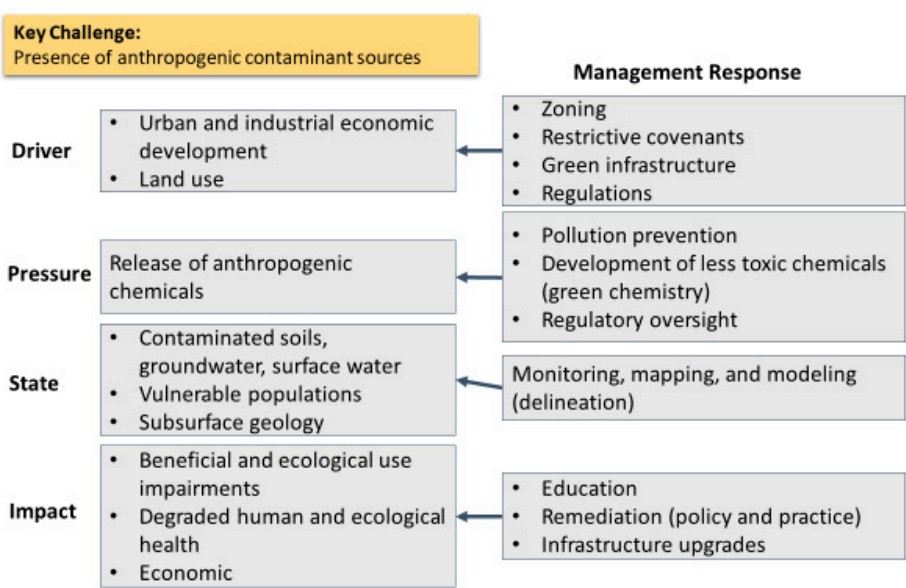

**Figure 5.** Anthropogenic Contaminant Sources Model.

Management responses to the drivers focus on future development and land use constraints. These include changes to zoning, issuing restrictive covenants to manage land use, utilizing green infrastructure to manage stormwater, and stringent regulations. Management responses to the pressure include pollution prevention, using alternative chemicals (e.g., green chemistry), and additional regulatory oversight. In terms of managing the state of contaminants in urban groundwater, substantially more groundwater quality data are needed. Additional monitoring, mapping, and modeling can all play a role in determining the location and spatial extent of groundwater contamination. It is also critical to understand the subsurface geology of these urban areas so that permeable vs. impermeable areas can be differentiated. Management responses to impacts include educa-

tion on urban pollutants, re-examining current remediation strategies, and infrastructure upgrades. Remedial actions include clean-ups at contamination sites and implementation of policies that encourage brownfield redevelopment.

Groundwater Table Model

Climate variability and changes to drainages drive changes to urban groundwater tables (Figure 6) [96]. The pressure is the change in water recharge and discharge. This creates a new state of elevated water tables; when the change exceeds certain thresholds, there can be increased vulnerability of neighborhoods and infrastructure. The impact is increased flooding, stormwater management issues (e.g., combined stormwater overflows), mobilizing contaminants, and adverse impacts to the economy [97–99], including disproportionate economic and social impacts to low-income communities.

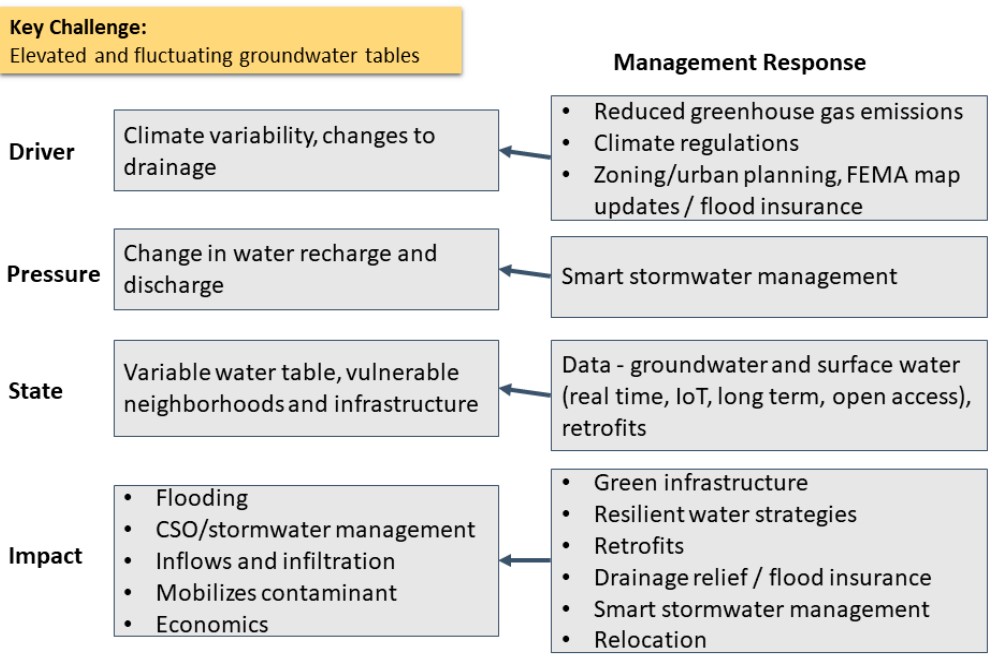

**Figure 6.** Elevated and fluctuating groundwater tables model.

Management responses to the drivers focus on managing climate change through climate regulations and reduced greenhouse gas emissions and zoning, which can include urban planning, FEMA map updates, and flood insurance management. The management responses to the pressure, state, and impacts all include similar approaches. These include the use of new smart or interconnected monitoring systems such as IoT (Internet of Things) infrastructure, edge computing or additional large data management systems, retrofits to existing systems or new GSI, and the need for better data management systems. New resilient water strategies, drainage relief, or potential relocation are additional management responses to the impacts of elevated and fluctuating groundwater tables.

Anthropogenic Modifications Model

Drivers to anthropogenic modifications of urban groundwater systems include urbanization, industrialization, impervious surfaces, aging infrastructure, and lack of data transparency (Figure 7). Pressures include further expansion of impervious surfaces and changes to surface and groundwater flows. The state is reduced infiltration, increased peak stream flows, and altered groundwater flow paths. Impacts include accelerated contaminant transport through changes to groundwater flow regionally and locally, modifications to the natural hydrologic cycle, and degraded groundwater and surface water quality.

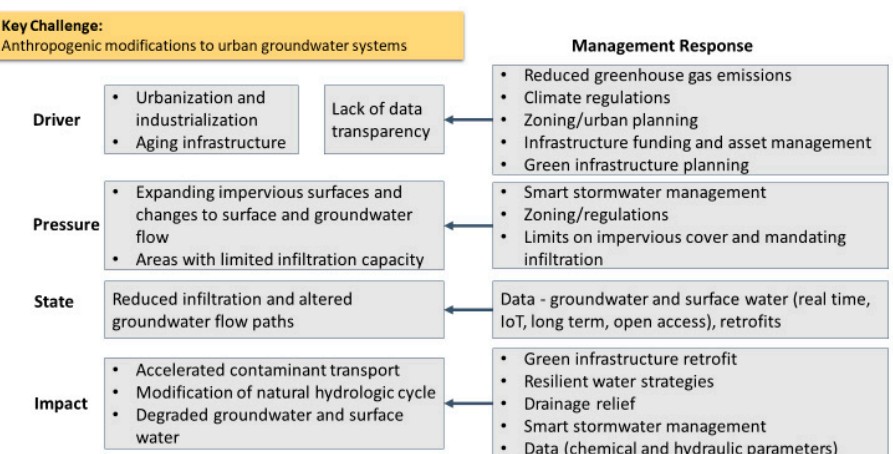

**Figure 7.** Anthropogenic modifications to urban groundwater systems model.

The management responses in the anthropogenic modifications model were similar to those in the Groundwater Table Model and include highlighting the need to address climate impact to groundwater, additional data management techniques such as smart (e.g., gated and storage-mediated) systems, and adjustments to existing water management systems (retrofits and GSI). As storm events increase in intensity, real and consequential risk mounts on surface water systems (i.e., flow and stream power increases) if groundwater systems cannot be relied on to reduce some of that periodic influx. The restoration, design and placement, and maintenance of large-scale, system-wide groundwater infiltration capacity will be a critical part of hydrologic planning in the urban landscape over the next several decades as climate impacts increase; actions may include providing additional storm drain capacity, replacement of impermeable with permeable surfaces, and installing more resilient infrastructure. Importantly, the design of any of these interventions must be informed by both future climate change predictions, as non-stationarity must be recognized, as well as understanding the surficial and near-surficial geology, to ensure these areas can accept infiltration.

### 3.3.3. Discussion

To address groundwater challenges in the urban sector, three categories of potential activities were identified: (1) Policy and Practice: increase public interest, and lobbying, zoning, and conservation activities; encourage native vegetation in landscaping; (2) Science/information gaps/infrastructure: identify data gaps and plans to develop more accessible data; increase consideration of environmental justice; and (3) Education and Outreach: collaboration and cooperation among education and outreach action lists; generate a media strategy using social media; create story maps. Below, the actionable next steps are listed for each category.

Policy and Practice

- Address urban land use concerns: zoning, restrictive covenants, regulations
- Increase the use of green infrastructure to manage urban water
- Prevent anthropogenic pollutant releases in urban areas: pollution prevention, development of less toxic chemicals (green chemistry), regulatory oversight
- Manage urban water in a changing climate: climate regulations, zoning, urban planning, FEMA map updates, flood insurance, infrastructure funding and asset management, green infrastructure planning

Initial implementation of the next steps includes the following: (1) create interest with voters/interest groups, including the targeting of membership organizations and professional societies and the Michigan Infrastructure Council; (2) identify funding sources (e.g., local bonds, linked with green and blue impact bonds and in sustainability-linked

(e.g., ESG [Environmental, Social, and Governance]) investments in the municipal markets (SEC Reg D and Reg CF securities); and (3) lobby (e.g., state and federal committees that control science, agriculture, and natural resources budgets).

Science/Info Gaps/Infrastructure

- Need for better stormwater management in urban centers: green infrastructure, resilient water strategies, retrofits, drainage relief, smart stormwater management, relocation, flood insurance.
- Address urban land use concerns: zoning/regulations, limits on impervious cover and mandating infiltration where feasible, alternative de-icers to minimize salinization of groundwater and surface water.
- Data—groundwater and surface water (real-time, IOT, long-term, open-access), retrofits.

Initial implementation of next steps includes: (1) create a data inventory, which will help identify the current data gaps in urban groundwater; (2) develop a data management system—standardize electronic data deliverables in Great Lakes cities; and (3) develop vulnerability/resilience indices.

Education and Outreach

- Education ideas:
  - Pollution prevention, development of less toxic chemicals (green chemistry)
  - Stormwater management
  - Salinization reduction
  - Infrastructure upgrades
  - Climate impacts
  - Scientific process

- Outreach ideas:
  - Green infrastructure
  - Flood insurance
  - Citizen science
  - Environmental justice
  - Data translation
  - Conservation
  - Native plants/grasses

Suggestions for initial implementation include: (1) develop a communication strategy for outreach using communication professionals and story tellers; and (2) create a media strategy and a list of key talking points, and generate a series of story maps.

### 3.4. Groundwater in the Coastal Wetland Sector

Coastal wetlands are often located at the interface of surface water and groundwater, representing systems of both groundwater discharge and recharge, sometimes with the same location serving both functions, depending on the season and hydrogeomorphic setting of the wetland [20]. Though groundwater flows can move in both directions, groundwater discharge is dominant. In coastal areas of the Great Lakes basin, groundwater discharge is present throughout the year, whereas in the lake beds, groundwater discharge dominates in the winter and recharge in the summer [21]. The coasts of Michigan harbor the most coastal wetlands of any Great Lakes state or province [100]. Much of the northern lower Michigan coast has high potential for groundwater discharge along the eastern coast of Lake Michigan and the northwestern coast of northern Lake Huron, related to the higher permeability of the glacial till and outwash deposits compared with other areas of shoreline [21,101] and to the height of the groundwater table relative to lake water levels [16]. These geologic conditions indicate that groundwater inputs can impact coastal wetland extent and ecosystem processes throughout the year.

3.4.1. Key Challenges

The coastal wetland breakout group identified three key challenges associated with groundwater in coastal areas: (1) climate change, (2) development of coastal areas, and (3) competing human and environmental uses for groundwater resources.

Climate change—coastal areas in the Great Lakes region are not only adapted to but depend on temporal variation (seasonal, annual, decadal) in water levels associated with weather and climate [102] for their ecological health; however, anthropogenic climate change has altered the natural variability leading to extreme high and low water levels across the region over relatively short periods [103,104]. Water levels in the Great Lakes can influence groundwater movement into coastal areas via differences in hydraulic head [20,105]. Subsequent changes in the direction of groundwater movement or the amount of discharge can potentially alter the source water and thus the water chemistry and habitat conditions of coastal wetlands [106]. In addition to changing hydrologic regimes, climate change may alter patterns of groundwater extraction for human uses, further impacting groundwater inputs to coastal wetlands. For example, warming temperatures associated with climate change are pushing agricultural land use, and the associated water withdrawals, toward northern Michigan [38,42,43,107], potentially decreasing groundwater levels and thus altering inputs to coastal areas of northern Michigan. Though unrelated to climate change or agriculture, groundwater drawdowns in an aquifer near Lake Erie caused decreases in groundwater discharge into coastal areas, resulting in downwelling of coastal lake waters into the sediments [106].

Development—coastal areas in Michigan are highly desirable for human development, and the abundant supply of fresh water has fueled development in the region [16,108]. This development has already contributed to a loss of approximately 50% of coastal wetlands in Michigan overall and over 90% in the Saginaw Bay area [109]. Despite this, Michigan still contains the largest extent of coastal wetland area across the Great Lakes region [100]. Development includes various land uses, such as residential, commercial, and industrial properties, as well as roads and green spaces. Frequently, impervious surfaces associated with developed areas reduce infiltration, thus decreasing groundwater recharge and increasing surface runoff [110,111]. The consequence of this could be altered sources, quantity, timing, and quality of water fluxes to coastal wetlands [112]. Additionally, the specific design of stormwater infrastructure and underlying conditions (soils, aquifer depth) can result in development either decreasing or increasing groundwater recharge [113]. Therefore, predicting the effects of development on groundwater recharge may vary from community to community and thus wetland to wetland, based on surficial geology. When groundwater recharge is reduced, there is potential that groundwater flows will also decline, decreasing groundwater inputs into coastal wetlands [114], which may in turn influence the extent and water chemistry of coastal wetlands [106]. Additionally, flooding related to climate-influenced high water levels is exacerbated by development and impervious surfaces in coastal areas, so high lake water levels in developed coastal areas could have a greater effect on water quantity and quality in coastal wetlands, relative to groundwater.

Competing human and ecological uses of groundwater—surface water is an abundant resource in Michigan, and Michiganders are broadly aware of its importance as a resource across the state, particularly as it relates to recreation, tourism, and industry. This awareness does not always extend to groundwater, which can contribute to competing human and ecological uses of groundwater. One example of this relationship in coastal areas is septic systems and groundwater. Humans extract groundwater for residential uses, and then residential wastewater is removed from homes through septic systems. Septic systems, however, are notoriously leaky, which can result in contamination in coastal environments through groundwater flow paths [115–117]. Industry and agriculture remove large quantities of groundwater, possibly reducing groundwater flow into coastal areas, and thereby altering critical habitat for fish production. Across Michigan, people are aware of the importance of the recreational fishing industry; however, many people are likely not aware

of the important role groundwater plays in maintaining healthy coastal wetland habitats for fish production.

### 3.4.2. DPSIR Models
### Climate Change Model

The driver for this model is climate change. We identified three pressures and their subsequent state changes that occur in coastal wetlands (Figure 8).

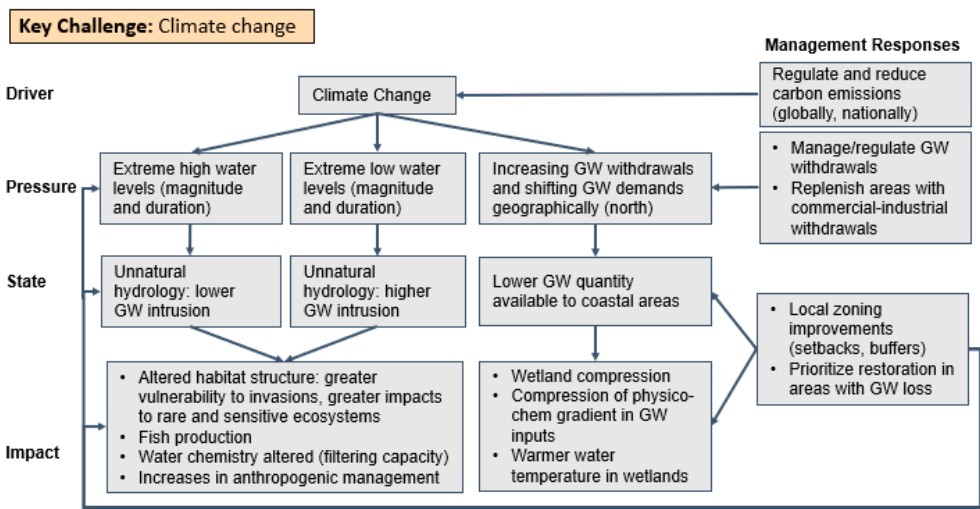

**Figure 8.** Climate change model.

(1 + 2) Extreme high water levels (1) and extreme low water levels (2), both in magnitude and duration, can cause unnatural hydrology in coastal areas. These pressures are two sides of the same coin and thus are discussed together below. These changes in water levels can change the hydraulic pressure at the surface water–groundwater interface, and result in lesser (under high water conditions) or greater (under low water conditions) groundwater discharges into coastal wetlands [21]. When lake water levels are high (as the Great Lakes were in 2017–2020), groundwater inputs may be restricted due to greater pressure from the higher surface water, decreasing the input of cooler, higher alkalinity groundwater into coastal wetlands. The opposite occurs during low water levels (as occurred in 2011–2013); notably, both change the physiochemical conditions of coastal wetlands.

These variable hydrologic states and shifts in groundwater inputs can result in changes to wetland extent and habitat structure (i.e., vegetation) with potentially greater impacts to vulnerable wetland vegetation communities, such as wet meadow marshes [118]. Changes to habitat structure, in addition to changes in water chemistry associated with fluctuations in groundwater discharge, in turn can influence the fish and macroinvertebrate communities that utilize the wetlands [106]. It is also possible that one of the primary services of coastal wetlands—their ability to filter nutrients and pollutants entering the lakes—may be diminished if wetland extent or vegetation structure is reduced. Increased phosphorus levels in the Florida Everglades have resulted in the replacement of native sawgrass (*Cladium*) with cattail (*Typha*), thereby disrupting and negatively impacting ecosystem function [119]. Anthropogenic attempts at management for high and low water levels in the Great Lakes, through the installation of seawalls and dredging activities [120], may further diminish connectivity to groundwater. This issue is exacerbated by the combination of natural variation and climate-change-related variation in water levels, which can be hard to tease apart and thus makes predictions difficult.

(3) The third climate change pressure is related to the trend of increasing groundwater withdrawals both throughout the state of Michigan and especially in Michigan's northern Lower Peninsula. This may be due to an increased demand for irrigation and a northward movement of agricultural land use and increased demand for irrigation [38,42,43,107].

This could result in a new state of reduced groundwater inputs to coastal areas as more groundwater is extracted for agriculture irrigation, although this may be offset by the thick glacial aquifers in coastal areas. An impact of decreased groundwater inputs is the compression of coastal wetlands; the vegetation communities nearest the land (i.e., wetlands, forest-shrub, and interdunal wetlands) would suffer the greatest losses. Rare and sensitive wetland ecosystems, like lake-level-dependent interdunal wetlands, are most at risk from decreases in groundwater inputs because groundwater is a major component of their source water, as opposed to other coastal wetlands which receive inputs from the adjacent lakes [20,121]. The compression of the wetlands subsequently compresses the water chemistry gradient typically present in coastal wetlands and ultimately changes both the habitat available to fish communities for production and the wetland extent providing ecosystem functions such as nutrient reduction.

The management responses vary in scale from drivers to impacts. Responses to the threat of climate change involve regulation and reduction of carbon emissions, which is beyond the scope of this summit. To address the pressure associated with groundwater withdrawals, the state could manage and regulate groundwater withdrawals, to protect and maintain groundwater flows to coastal areas in a similar manner to the one applied to streams and rivers in the Water Withdrawal Assessment Tool [122]. Regulations could also be put in place to replenish groundwater reservoirs from which commercial and industrial withdrawals take place. Improved local zoning regulations to protect wetlands (through setbacks and buffers) would address the pressure, state change, and impacts associated with variable water levels, as well as the impacts associated with increased groundwater withdrawals. A second management response would be to prioritize restoration efforts on coastal wetlands most impacted by water level variability and wetlands with greater potential for groundwater inputs.

Development Model

The DPSIR model for the key challenge of human development is related to land use changes that result in greater amounts of impervious surfaces as a primary driver of change to coastal wetlands via the pressure of reduced connectivity to groundwater (Figure 9; see also the urban section). Reduced connectivity between surface waters and groundwater has a two-part state change. First, development and the associated increase in impervious surface area decrease infiltration into groundwater, which results in a lower groundwater table making groundwater less available to coastal areas [110,111]. Second, there is less groundwater discharge into coastal areas because development/impervious surfaces promote runoff while reducing recharge, and thereby restrict groundwater inputs into surface waters. The impacts of these state changes result in similar coastal wetland habitat impacts described in the climate change section above.

Management responses to mitigate impacts of impervious surfaces could include regulatory requirements for developed areas to include green infrastructure and stormwater management options that maintain groundwater recharge. To address the pressure of reduced connectivity associated with development, there is an ongoing need to implement green infrastructure, increase stormwater infiltration into groundwater, and remove structures that inhibit groundwater flows into existing developed coastal areas. To address the change in state and impacts of development, management could focus on local zoning improvements, such as setbacks and buffers along coastal wetlands. There also could be prioritization for restoration efforts in coastal areas where high groundwater inputs have been lost to development.

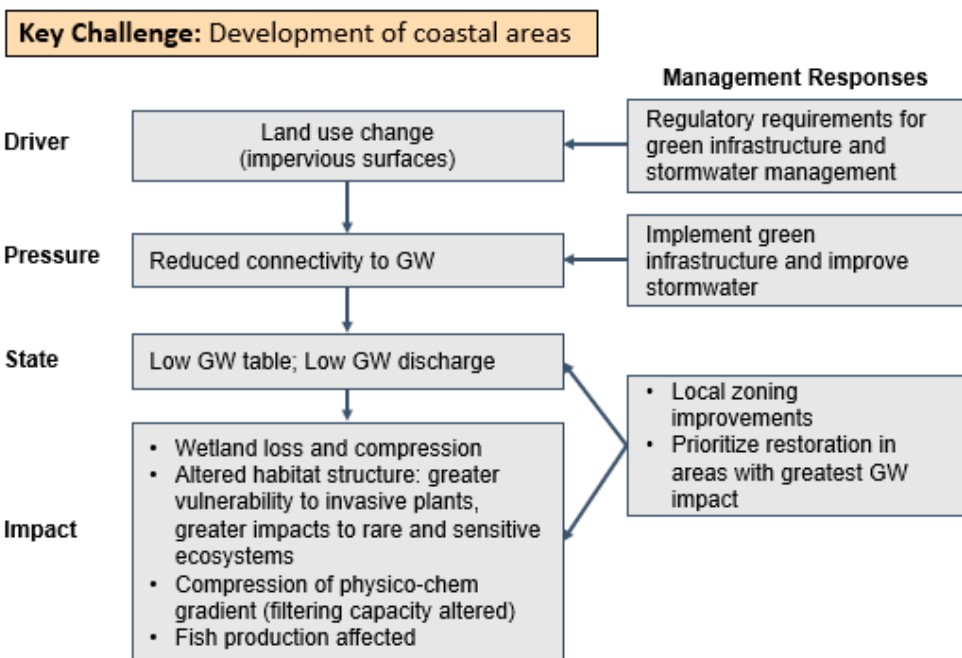

**Figure 9.** Development of coastal areas model.

Competing Human and Ecological Uses of Groundwater Model

A generalized DPSIR model was developed for the challenge of competing human and ecological uses of groundwater, which focused on how and why research on this topic should be conducted (Figure 10). This model has a different format, with a focus on where science and research can contribute to the challenge. The driver describes an overall low societal awareness of groundwater issues and the importance of groundwater as both a resource and for its connectivity to surface waters. The pressure is the overall lack of research with a hydrogeology focus in Great Lakes coastal wetlands, as most researchers focus on the surface biology of these systems. There is also a lack of trigger events, which would put greater emphasis on groundwater-related issues. The overall state is that we lack a science strategy for understanding groundwater inputs and we have an incomplete picture of this issue. We subsequently lack a management strategy, with the resulting impact that decisions on groundwater may be inadequately formed, which jeopardizes our ability to protect coastal wetland ecosystems.

The management responses to this challenge primarily involve acquiring more knowledge on groundwater influences in coastal areas, in order to make informed management decisions. The first step would be to fund the development of groundwater strategic programming, as well as associated research projects and infrastructure to monitor groundwater flows. Ideally, there would be funding for groundwater monitoring and modeling, as proposed in the Michigan Hydrologic Framework (discussed below), as well as a groundwater resource research focus at national laboratories. This groundwater research focus should extend beyond remediation (e.g., Strategic Environmental Research and Development Program—SERDP, Environmental Security Technology Certification Program—ESTCP) and monitoring (United States Geological Survey—USGS). Assessing the contributions of groundwater flows to surface ecosystems and evaluating the effects of changes in groundwater supply and discharge to surface waters are recommended. Groundwater monitoring infrastructure could be installed at sites with ongoing research and data collection infrastructure. A comprehensive groundwater data management system would also be useful for researchers and managers, as it would link research to management programs and ultimately be used for groundwater decision-making. For the public, funding for education on groundwater through outreach or courses will be important to bridge the gap in understanding of groundwater issues.

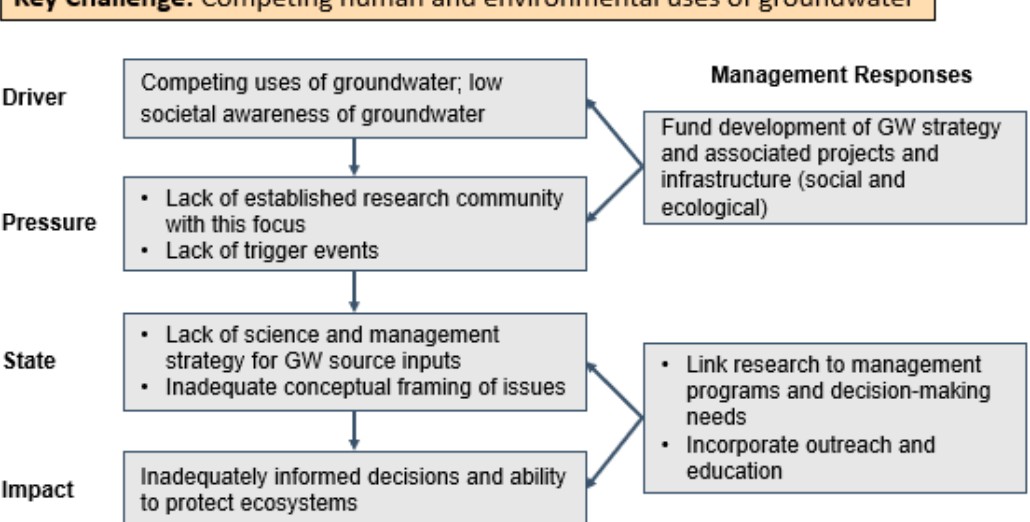

**Figure 10.** Competing human and environmental uses of groundwater model.

### 3.4.3. Discussion

The broad challenge related to groundwater in coastal areas, across both research and management, is the limited knowledge about groundwater, as described in Figure 10. The jurisdiction for groundwater research in the funding community is ambiguous and undefined, making research on the topic more complicated and harder to accomplish. Though the USGS has a research focus on groundwater, their granting programs require a match (unlike other research area granting programs) and they are not allowed to make management recommendations; hence, not only are funding opportunities limited, but the scientists themselves are constrained to a certain degree. Groundwater research and management at the state level would greatly benefit from more federal support. Additionally, we acknowledge the need to include more groups in groundwater management, as mentioned previously, including governmental agencies, Indigenous communities, and non-governmental organizations.

To move forward with addressing groundwater concerns in coastal wetlands, several key actionable next steps are recommended related to policy, science, and education. Though these action items were identified as the most urgent, all of the management responses listed in the DIPSR models are important for the long-term management of groundwater inputs in coastal wetlands.

Policy and Practice—to address the impacts of both the development and climate change challenges in coastal wetlands, zoning improvements, such as setbacks and riparian buffers, were identified as important policy items to be implemented. The DPSIR models for the climate change and development drivers resulted in similar impacts to coastal wetlands, thus local management responses are similar, and solutions for one challenge also support solutions for the other challenge. The zoning policies would be strongest if statewide standards could be set across Michigan (and other states). The group also recognized the need for greater groundwater prioritization by the state (for policy, research, and management), and noted that more funding is needed for the Michigan Department of Environment, Great Lakes, and Energy (EGLE) in order to implement such programs.

Science and Infrastructure—in order to make informed policy and regulatory decisions, there is an urgent need in the scientific community for more information on: (1) the effects of groundwater withdrawals on coastal wetland water quantity and quality, and (2) the overall connectivity of groundwater and surface waters in coastal wetlands. The source waters for Great Lakes coastal wetlands are more complicated than palustrine (inland) wetlands, as coastal wetlands have water contributions from the Great Lakes themselves, riverine systems, and groundwater (through varying subsurface flow paths). Each of

these water sources has a unique chemistry, which, when combined, defines the wetland chemistry (e.g., alkaline fens vs. acidic bogs). This, in turn, ultimately influences the habitat conditions for biological communities. Increasing groundwater monitoring infrastructure and modeling capabilities in coastal areas with varying land uses would be an important initial step to address the lack of knowledge in this sector.

Education and Outreach—to increase education on groundwater issues in Michigan, partnering with Michigan Sea Grant and its Extension programming is suggested. Groundwater researchers and managers could work with Sea Grant and Extension agents to seek sustained funding for a groundwater education and outreach program, which could be an independent program or combined with existing programming. Funding for groundwater outreach and education could be sought through the Great Lakes Restoration Initiative Area #5, Foundations for Future Restoration Actions, which involves youth education and experiential learning opportunities.

## 4. Summary and Recommendations

Groundwater is a natural resource in peril, in Michigan and throughout the world. This likely is because we cannot see it, we do not measure its stocks and flows in a coordinated and consistent manner, and we have done a poor job of communicating its value to society at large. In this paper, we attempt to address the key challenges facing groundwater in Michigan, with the intent that the information generated can be transferable to the Great Lakes region. We recognize (and hope) that our summit outcomes have broader application than just Michigan and the Great Lakes basin. Indeed, there are clear applications to the UN sustainable development goals [123], as water availability links directly or indirectly to virtually every goal. Specifically, SDG 6 (clean water and sanitation) and 14 (life below water) are the most direct links, but SDG 1 (no poverty), 2 (zero hunger), 3 (good health and well-being), 10 (reduced inequality), 11 (sustainable cities and communities), 12 (responsible consumption and production), 13 (climate action), and 15 (life on land) arguably are linked to groundwater quantity and quality.

We specifically targeted three sectors and utilized a conceptual modeling framework to make our results more visually intuitive. Unlike many studies that have addressed groundwater resources from broad (e.g., over-withdrawal) or contaminant-specific (e.g., PFAS) approaches [1,2,124], our approach allowed us to identify a full range of groundwater-related issues in critical systems, some with significant socio-economic implications (agriculture and urban systems) and another representing a valuable natural ecosystem [125,126]. Our conceptual models were intentionally designed to identify information gaps but also provide specific recommendations that resource managers can implement to protect, restore, or enhance the services and functions provided in these sectors.

The agricultural work group emphasized the challenges associated with irrigation, contaminants, and best management practices to address climate change. The urban work group focused on fluctuating groundwater tables, anthropogenic modifications to the groundwater system, and, as with agriculture, contaminants. Finally, the coastal wetland work group identified the key challenges of climate change, development, and competition between humans and the environment for groundwater. The conceptual models helped visualize the relationships and showed the specific management recommendations for each challenge. A number of recommendations transcended sector boundaries, and are listed below.

Scientific Recommendations:

- Develop a statewide groundwater budget
- Coordinate data collection/management activities into a coordinated information management system
- Enhance and refine the Michigan Water Use Program and the Water Withdrawal Assessment Tool
- Develop a statewide groundwater monitoring program focused on contaminants
- Develop an early warning system to envision the future state of supply and demand

- Develop an interactive decision-making tool to quantify the impact of potential new withdrawals based on real-time groundwater monitoring data and enhanced geologic mapping data

  Management-oriented Recommendations:

- Improve our public education and outreach efforts to improve the public's general lack of understanding of groundwater, and especially its connectivity to surface water
- Create new information and visualization tools to explain groundwater science and policy
- Instill the importance of water conservation
- Garner more input from underrepresented communities to obtain multiple perspectives
- Although we did not reach a consensus on how we should advocate on behalf of groundwater as a resource, there was general agreement regarding the need for more effective strategies to garner the resources and attention on groundwater as a growing Great Lakes issue
- The Michigan WUAC is statutorily charged to report and make recommendations biennially to the Legislature. It can be an effective advocate for groundwater in Michigan given its diverse membership, with appointees representing: business and manufacturing, public utilities, anglers, agricultural and non-agricultural irrigators, well drillers, local units of government, wetlands conservation, municipal water supplies, riparian landowners, professional hydrogeologists, Indian tribes, the aggregate industry, environmental organizations, and local watershed councils.

Our groundwater summit was not without its limitations. The summit lacked representation from the BIPOC community; it is clear that their input needs to be included in the future to recognize and affirm the value, existence, and validity of knowledge and information from all sources. In addition, although our approach allowed us to evaluate groundwater from multiple perspectives within three different sectors, we did not attempt to look at how groundwater resources may be impacted by interactions among these sectors. This may be a rich topic for the future, perhaps requiring multi-objective optimization processes [127].

The increasing attention being placed on groundwater is long overdue but also is reactive in nature, stemming from conflicts over water quantity and public health warnings over water quality. This is a poor way to manage such a critical natural resource. Society must become better informed about this resource and be more proactive in managing groundwater. There are many efforts currently underway both in Michigan and across the Great Lakes, including the Michigan Hydrologic Framework; Michigan Water Use Advisory Council; Michigan Groundwater Table; a groundwater governance in the Great Lakes region study; an International Joint Commission groundwater–surface water conceptual framework study; a USGS study focused on Great Lakes hydrologic and hydrodynamic models and data sets; a Detroit regional groundwater study; and Ottawa County's (MI) Groundwater Evaluation and Response Coordination System. These efforts are indicative of the interest and need to better understand groundwater in the region, and we encourage these initiatives to coordinate with one another to optimize their activities and avoid redundancies.

With more coordinated information management, better understanding of groundwater stocks and flows, and improved education and outreach, we can move from a reactive management model to a proactive one regarding Michigan's groundwater resources. These actions will help transform the sustainability of this critical resource from imperiled to secure.

**Supplementary Materials:** The following supporting information can be downloaded at: https://www.mdpi.com/article/10.3390/su14053008/s1, Table S1: Breakout Group Assignments; Table S2: Summit Agenda.

**Author Contributions:** Conceptualization, A.D.S.; methodology, A.D.S., D.G.U., P.D., T.Z., C.M. and P.C.; formal analysis, A.D.S., D.G.U., D.P.L., C.M., P.D, T.Z., P.C., J.A. (Jon Allan), J.A. (Jeremiah Asher), J.B., D.C., D.D., C.D., J.E., A.G., A.H., L.D.L., J.N., W.O., B.O., P.S. (Paul Sachs), P.S. (Paul Seelbach), T.S., A.S., J.Y.; investigation, all authors; writing—original draft preparation, A.D.S., D.G.U., D.P.L.,

C.M. and P.D.; writing—all authors; project administration, A.D.S.; funding acquisition, A.D.S. All authors have read and agreed to the published version of the manuscript.

**Funding:** This summit was funded by the Cooperative Institute for Great Lakes Research (CIGLR) at the University of Michigan and National Oceanic and Atmospheric Administration and the Allen and Helen Hunting Research and Innovation Fund held at the Annis Water Resources Institute at Grand Valley State University, grant number NA17OAR4320152.

**Acknowledgments:** Logistical support was provided by Emily Kindervater (AWRI-GVSU, now at Hope College) and Mary Ogdahl and Tom Johengen from CIGLR, as well as Emilia Ferme Giralt and John Bratton (LimnoTech). We also express our gratitude to Lauren Fry (NOAA-GLERL), Ralph Haefner (USGS), and Mindy Erickson (USGS), who were members of the Summit. Finally, the lead author expresses his appreciation to three anonymous reviewers for their thoughtful and constructive comments.

**Conflicts of Interest:** The authors declare no conflict of interest.

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
