# Peer review of "Groundwater in Crisis? Addressing Groundwater Challenges in Michigan (USA) as a Template for the Great Lakes"

_sustainability, doi:10.3390/su14053008_

Round 1
Reviewer 1 Report
This study is interesting and the methods used are effective. However, the current manuscript still contains some flaws and incomprehensible expressions. Therefore, I propose a moderate revision of the manuscript with the following comments.
- I suggest focusing on checking some typo errors to make it easy to understand for the readership of the journal.
- I strongly advise the authors to improve Introduction and Discussion according to suggested articles. These research articles have identified related topics of sustainable water management. I believe it will improve the quality of your work. I strongly suggested authors improve this section a bit more. I advise authors to revisit their literature section of the recommended studies
[1] Managing aquifer recharge with multi-source water to realize sustainable management of groundwater resources in Jinan, China
- The shortcomings of previous studies and the innovation and motivation of this study should be emphasized in the Introduction.
- It is suggested to present the structure of the article at the end of the introduction.
- It is suggested to add the limitations of this study in the discussion section. Please refer to the following literature.
[2] Assessment of river health based on a novel multidimensional similarity cloud model in the Lhasa River, Qinghai-Tibet Plateau
- It is suggested that a separate subsection be added to the discussion section to link the significance of this study to the UN sustainable development goals (SDGs).
- I want to see this creative work after some corrections. I have endorsed this study as; it deserves the merit for publication. However, I suggest the authors make corrections according to my advice. Please read the suggested studies and execute them in the introduction and literature sections.
Reviewer 2 Report
Thanks for providing me an opportunity to review the entitled manuscript"Groundwater in Crisis? Addressing Groundwater Challenges in Michigan as a template for the Great Lakes". It's quite an interesting study that bridges the wide perspectives of inventory the key (grand) challenges facing groundwater in Michigan, identify the knowledge gaps and scientific needs, as well as policy recommendations, associated with these challenges, construct a set of conceptual models that elucidate these challenges, and develop a list of (tractable) next steps that can be taken to address these challenges. I recommend this work, however, authors should address some of my comments before publication. The author should address the following of my comments.
1- It is recommended that the author rewrite the Introduction, increase the citation of the literature, and extract questions and useful information from the literature. Through literature review, point out the shortcomings of existing research, thus leading to the article's environmental significance and purpose. In this section, the literature review needs to be more critical.
The authors should detail the methodological novelties with the vast amount of existing literature in this area.
2- What are the basis of technical and non-technical issues? I mean threshold to define them? It would be better if the author can summarize some details to clear the basis of defining technical and non-technical issues. Otherwise, land and water management is a technical issue in my opinion which is described as non-technical by authors.
3-Discussions should be expanded with updated literature.
Overall, the manuscript is written well.
Reviewer 3 Report
Please find the attached file.

Round 2
Reviewer 1 Report
The authors have addressed all my concerns and I agree to publish this paper.